# Mechanical Model and Seismic Response Analysis of a Track Type Combined Isolation Device

**Guanjie Kong, Ziyu Wang** and **Youfa Yang** *

College of Civil Engineering, Chongqing University, Chongqing 400045, China
* Correspondence: yfyang@cqu.edu.cn

**Abstract:** In view of the weak isolation and shock absorption effect of ordinary isolation bearings at present, a new track damping negative-stiffness device was designed based on the principle of negative stiffness. The principle of negative stiffness was applied to base isolation, and a new isolation system was proposed in which the track-type NSD device and the common isolation layer were connected in parallel. The track-type NSD had the characteristics of clear force transmission, simple structure, and self-resetting ability. The theoretical model of orbital NSD was established, and its hysteretic performance was simulated. The seismic response of a six-story reinforced-concrete-frame isolation structure system with a new track-type-damping negative-stiffness device was analyzed. The seismic responses of the traditional lead rubber bearing isolation model (LRB), the new track-type negative stiffness and lead rubber bearing combined isolation model (NSD), and the new track-type-damping negative stiffness and lead rubber bearing combined isolation model (DNSD) were compared. The results show that, in the same case, compared with the LRB model and NSD model, the DNSD model will further reduce the superstructure acceleration response and displacement response of the isolation system, and the greater the peak value of the input seismic wave, the more obvious the isolation effect of the structural system. The new track damping negative-stiffness device enhances the energy dissipation capacity of the structural system and plays a role in controlling the displacement of the isolation layer. The displacement response of the isolation layer and the ground motion response of the superstructure are reduced at the same time, and the isolation efficiency is improved.

**Keywords:** track type; negative stiffness; isolation device; seismic response reduction; isolation effect



## 1. Introduction

Base isolation technology is based on the use of a flexible connection between the superstructure and the foundation and sets up a safe isolation system to protect the superstructure. It has become one of the most effective means to reduce earthquake disasters. The effect of base isolation technology on long-period structures is poor. The reason is that isolation is mainly achieved by extending the natural vibration period and increasing damping. In order to make the isolation effect obvious (such as reducing 1 degree), the natural vibration period of the structure needs to be extended by more than two times. This makes it difficult to realize the isolation layer of long-period structures, and the displacement of the isolation layer may be large in rare earthquakes. Therefore, how to effectively extend the period of isolated structure, significantly increase damping, and control the displacement of isolation layer is the key problem of long-period structural isolation. The rubber bearing isolation system based on a negative-stiffness damping device (NSD) is one of the effective ways to solve these problems. Therefore, researchers introduced the negative-stiffness theory for the vibration isolation of mechanical equipment. By actively weakening the stiffness of some devices in the structure, the period of structural vibration was further extended to reach the "simulated yield" [1] state proposed by Pasala. The most representative is Iemura, who discovered and proposed the concept of "negative

stiffness" in the study of active variable damping control [2]. Iemura also introduced the negative-stiffness theory into civil engineering, established a numerical analysis algorithm, combined the developed negative-stiffness isolation device with ordinary rubber bearings, and verified the effectiveness of the negative-stiffness device in reducing structural displacement and acceleration through experiments [3]. Then, in the two-stage benchmark problem of cable-stayed bridges, Iemura proposed to compare the negative-stiffness control with active control and passive control, which proves the excellent performance of the pseudo negative-stiffness device [4]. Attary proposed a negative-stiffness device for seismic protection of highway bridge structures [5] and pointed out that the seismic response of bridge structures can be effectively reduced by paralleling the device with the structure and damping device, thus providing positive stiffness. In China, Yang Qiaorong et al. proposed a damping negative-stiffness device and studied its seismic response. The results show that the device cannot only reduce the absolute acceleration response of the superstructure under the action of long-term and short-term earthquakes, but it can also control the displacement response of the isolation layer and improve the isolation efficiency [6]. By applying the principle of negative stiffness to base isolation, Ji Han proposed a new isolation system in which the negative-stiffness damping device is connected in parallel with the ordinary isolation layer and carried out theoretical research on the system [7]. Li Xuan and others designed a new friction sliding composite isolation system and analyzed it with finite element software. It was concluded that the friction sliding bearing with a small amount of thick rubber bearings and viscous dampers can effectively reduce the residual deformation after the earthquake and the maximum displacement of the isolation layer [8]. Sun Tong et al. proposed a track-type negative-stiffness device and studied its damping control. The results showed that the displacement control effect of the device is equivalent to that of LQR semi-active control, and the acceleration response control effect is much better than that of LQR semi-active control [9].

Most of the existing negative-stiffness devices provide negative stiffness at the preset displacement, which is difficult to control the displacement of the isolation layer under different ground motions and reduce the response of the superstructure. Based on past devices [6], a new type of adaptive variable-stiffness isolation device is illustrated in this paper. Different from other devices, it integrates the flexible track concept and the concept of negative-stiffness damping system into the variable-stiffness device, and a variable-stiffness isolation system is proposed in combination with lead rubber bearings so that the stiffness of the isolation layer presents different stiffness characteristics under different displacement states. Thus, it can provide positive stiffness to limit the displacement of the isolation layer under interference load, provide negative stiffness under different seismic forces, control the displacement of the isolation layer, and achieve effective isolation, so as to meet the design requirements of different levels of seismic performance. It does not only realize the effective isolation of long-period structures but also improves the disadvantages of the previous negative-stiffness devices, such as their complex structure and residual deformation. In addition, the track surface function or other parameters can be flexibly changed according to the design requirements to meet the isolation requirements under different conditions.

## 2. Negative-Stiffness Device and Its Mechanical Model

### 2.1. Negative-Stiffness Device

Negative stiffness means that, in the force–displacement curve, the force decreases with the increase of displacement, making the slope of this section of the curve negative, as graphically illustrated in Reference [10]. Combined with the definition of stiffness, the concept of negative stiffness comes into being. In the traditional isolation layer, the sum of the horizontal stiffness of all rubber bearings is positive. If a negative-stiffness device is added to the isolation layer, the stiffness of the isolation layer can be further reduced (the total stiffness is still positive), so that its natural vibration frequency can be further reduced, thus effectively reducing the seismic response of the structure under

long-period earthquake. Based on the principle of variable stiffness and the idea of track type, a negative-stiffness device with damping is proposed which is mainly composed of a track plate, roller, preloaded Belleville spring and viscous damper in Belleville spring guide sleeve with the curve of plate surface as a function, as shown in Figure 1. Changing the function of the track surface curve can realize the flexible adjustment of the isolation performance of the device.

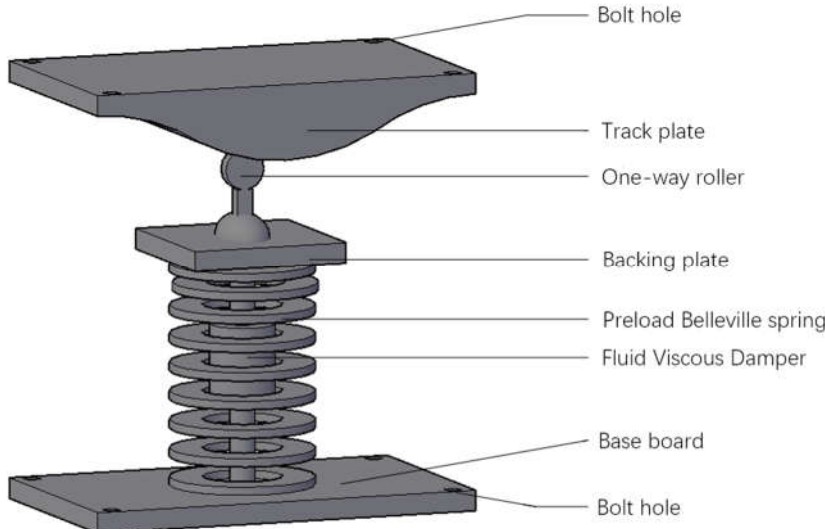

**Figure 1.** Schematic diagram of damping-track-type variable-stiffness device.

### 2.2. Mechanical Model

Set the initial spring length as $l_p$, the spring preload as $\Delta l$, the spring length after initial compression as $l_0$, the spring stiffness as $k$, and the friction coefficient of the track's surface as $\mu$. In the initial state, the position of the roller is in the coordinate system shown in Figure 2a; the function value of the track surface is $f(x_0)$, and the spring length at any time is $l$. With the horizontal displacement of the device, $u$, the function value of the track surface where the roller is located in the coordinate system, as shown in Figure 2b, is $f(x)$; the stress state of the corresponding roller is shown in Figure 3. At this time, the spring length is as follows:

$$l = l_0 + f(x) - f(x_0) \tag{1}$$

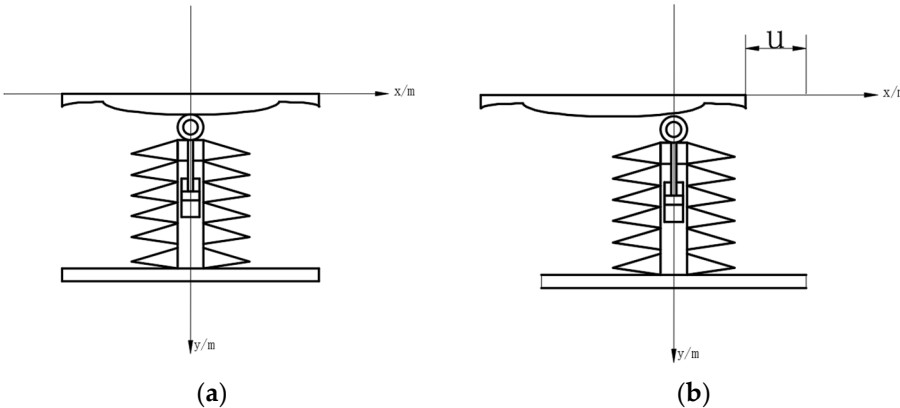

| (a) | (b) |

**Figure 2.** Motion state of negative-stiffness device. (**a**) Initial state of the device (**b**) Device state after horizontal displacement, u.

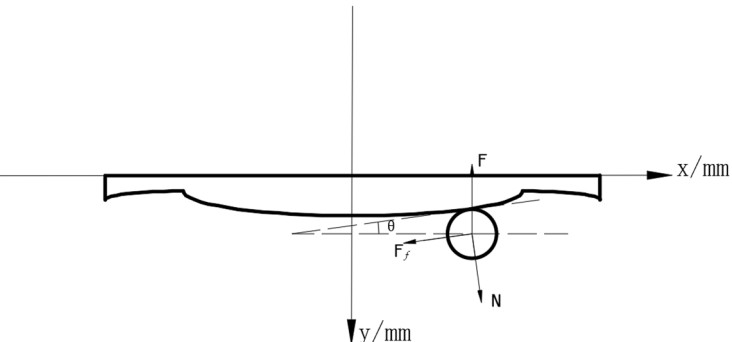

**Figure 3.** Stress-state analysis of roller.

The corresponding spring force, $F$, and the rail surface reaction, $N$, are calculated as follows:

$$F = k(l_p - l) \tag{2}$$

$$N = F \cos \theta \tag{3}$$

where $\theta$ is the angle between the tangent of the rail surface function at the roller position and the $x$-axis.

At this time, the friction force in the tangent direction of the track surface where the roller is located is as follows:

$$F_f = \mu F \cos \theta \tag{4}$$

Decompose $N$ and $F_f$ to the horizontal direction, and it can be concluded that the force provided by the spring part of the track-type-damping negative-stiffness device opposite to the displacement direction, $F_{NSDspring}$, is as follows:

$$F_{NSDspring} = F \cos \theta \sin \theta - \mu F \cos^2 \theta = k(l_p - l) \cos \theta \sin \theta - \mu F \cos^2 \theta \tag{5}$$

According to the conversion relationship between trigonometric functions, we have the following:

$$f'(x) = \tan \theta$$
$$\sin \theta \cos \theta = \frac{\tan \theta}{1 + \tan^2 \theta} \tag{6}$$
$$\cos^2 \theta = \frac{1}{1 + \tan^2 \theta}$$

It can be concluded that the formula for $F_{NSDspring}$, as expressed by the orbital surface function, is as follows:

$$F_{NSDspring} = k[\Delta l + f(x) - f(x_0)] \left[ \frac{f'(x)}{1 + f'^2(x)} - \mu \frac{1}{1 + f'^2(x)} \right] \tag{7}$$

Calculate the first derivative of the displacement $x$ in Formula (6) to obtain the relationship between the negative stiffness of the device and the displacement:

$$\begin{aligned} K_{NSD} &= kf'(x) \left[ \frac{f'(x)}{1 + [f'(x)]^2} - \mu \frac{1}{1 + [f'(x)]^2} \right] \\ &+ k[\Delta l + f(x) - f(x_0)] \left[ \frac{f''(x)[1 + f'^2(x)] - 2f'^2(x)}{[1 + f'^2(x)]^2} + \mu \frac{2f'(x)}{[1 + f'^2(x)]^2} \right] \end{aligned} \tag{8}$$

Let the damping coefficient of the velocity dependent viscous damper be $C$, and we can see that the initial stiffness takes 100 times of the damping coefficient $C$ [11]. Under the action of ground motion, with the horizontal displacement of the track plate, the relative position of the roller on the track surface changes, and this will produce a vertical velocity,

i.e., the axial velocity of the viscous damper, which is set as $v$. Then the relationship between $v$ and track surface function $f(x)$ is calculated as follows:

$$v = \frac{df(x)}{dt} = \frac{df(x)}{dx}\frac{dx}{dt} = f'(x) \cdot \dot{x} \tag{9}$$

Then the damping force generated by the viscous damper is decomposed to the horizontal direction as follows (the decomposition method is the same as the spring part):

$$\begin{aligned} F_{NSDdamper} &= c|v|^\partial \cdot \text{sgn}(\dot{x}) \\ &= c \cdot \text{sgn}(\dot{x}) \cdot |f'(x) \cdot \dot{x}|^\partial \left[ \frac{f'(x)}{1+f'^2(x)} - \text{sgn}(\dot{x})v\frac{1}{1+f'^2(x)} \right] \end{aligned} \tag{10}$$

In this paper, the commonly used $\alpha = 0.3$ is taken to discuss the mechanical properties of the device.

Since the spring providing negative stiffness in the device does not have an energy-dissipation capacity, additional viscous dampers are considered. Although additional damping can improve the energy dissipation capacity of the isolation layer, it will also increase the seismic response of the superstructure [6,12]. Therefore, it is necessary to comprehensively consider the seismic response of the superstructure and the displacement of the isolation layer to ensure that the set damping parameters are reasonable and effective.

In this paper, the form of the orbital surface function is the cosine function:

$$f(x) = b \cdot \cos(\beta x) \tag{11}$$

As $\beta = 10$, $b = 0.05$, $\Delta l = 0.5$, $u = 0.2$, and $k = 50$, the negative-stiffness mechanical model of the device can be drawn from Equation (6), as shown in Figure 4.

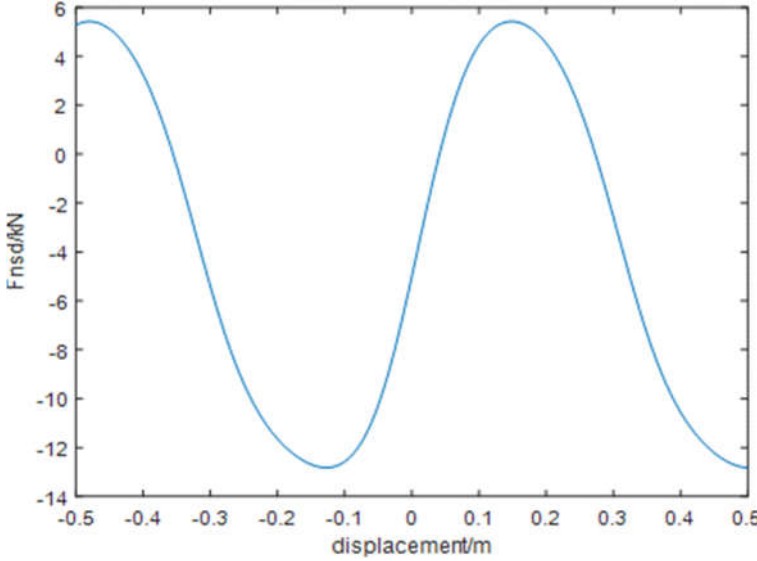

**Figure 4.** Mechanical model of negative−stiffness device.

The relationship between negative stiffness, $K_{NSD}$, and displacement, $x$, can be drawn from Equation (7), as shown in Figure 5.

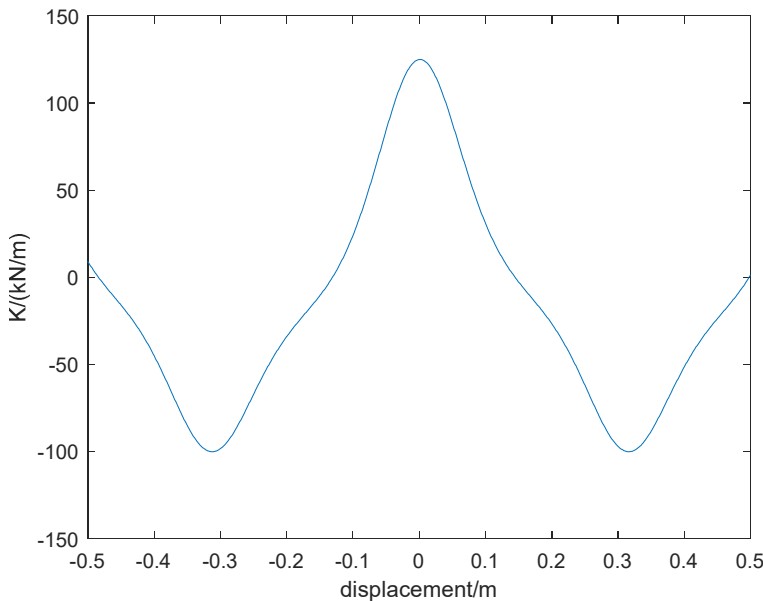

**Figure 5.** Relationship between $K_{NSD}$ and $x$.

The negative-stiffness section of the mechanical model of the device can be clearly observed from Figures 4 and 5.

The damping mechanical model of the device for different $\beta$ values can be drawn from Equation (9), as shown in Figures 6 and 7. $F_c$ is the horizontal force provided by the damper.

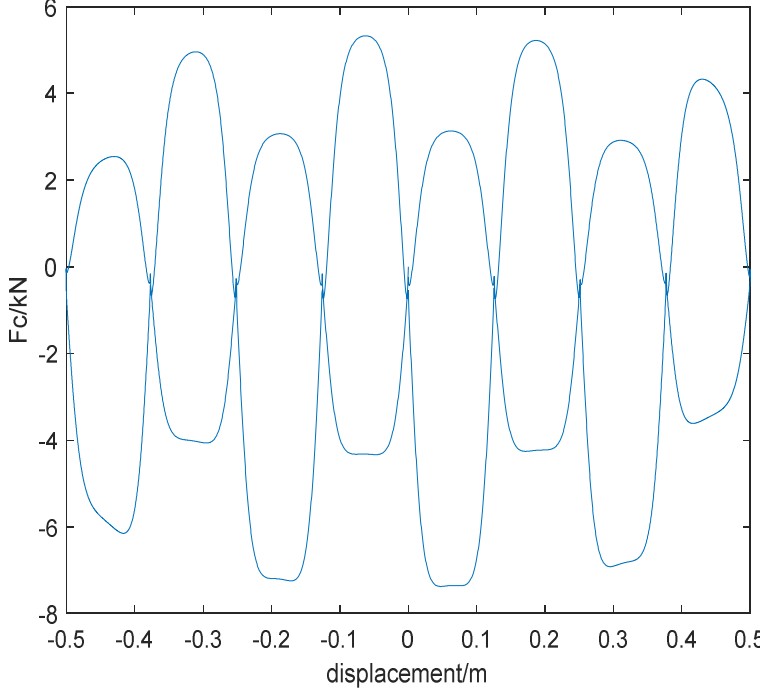

**Figure 6.** $C = 15$ and $\beta = 25$ damping mechanical model.

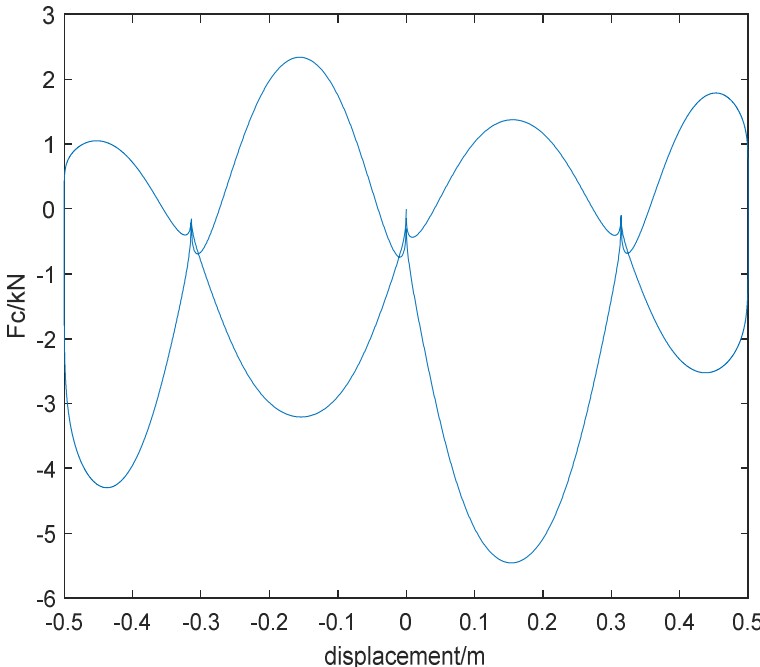

**Figure 7.** $C = 15$ and $\beta = 10$ damping mechanical model.

It can be seen from Figures 6 and 7 that the shape of the mechanical model of damping is basically unchanged, while the period and peak value of the model function change.

### 2.3. Parameter Impact Analysis

From the expression of $F_{NSDspring}$, the parameters that affect the mechanical properties of the device can be found. Using the control variable method, the frequency of orbital surface function frequency, $\beta$; the peak value of track surface function $b$; the stiffness of Belleville spring, $k$; the preload, $\Delta l$; and the friction coefficient of track surface $\mu$, are analyzed, respectively.

#### 2.3.1. Orbital Surface Function Frequency, $\beta$, and Peak Value of Orbital Surface Function, $b$, Influence on $F_{NSD} - x$ Curve

If we take $\Delta l = 0.5$, $k = 50$, and $\mu = 0.2$ from Equation (6), the force displacement curve under different $\beta$ values and the force displacement curve under different $b$ values can be obtained, as shown in Figures 8 and 9.

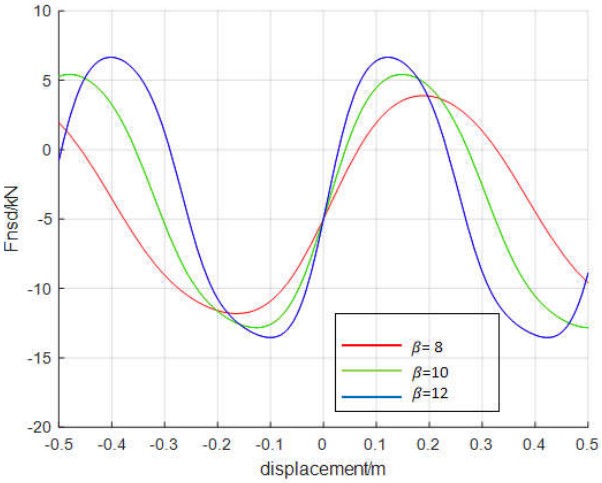

**Figure 8.** Force–displacement curve under different $\beta$ values.

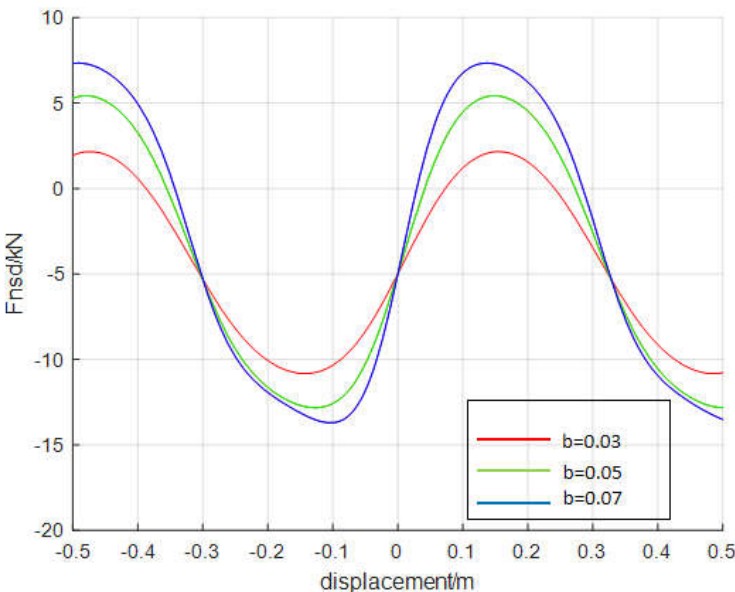

**Figure 9.** Force–displacement curve under different *b* values.

It can be seen from the above two figures that changing the value of $\beta$ will change the change cycle of force, and the stronger the value of $\beta$, the greater the change frequency; changing the *b* value of the track surface will change the magnitude of the force peak, but it will have no effect on the change frequency of the force.

### 2.3.2. Effect of Spring Stiffness *k* on Negative Stiffness

From Equation (7), the relationship between negative stiffness, $K_{NSD}$, and displacement, *x*, under different spring stiffness values, *k*, can be obtained, as shown in Figure 10.

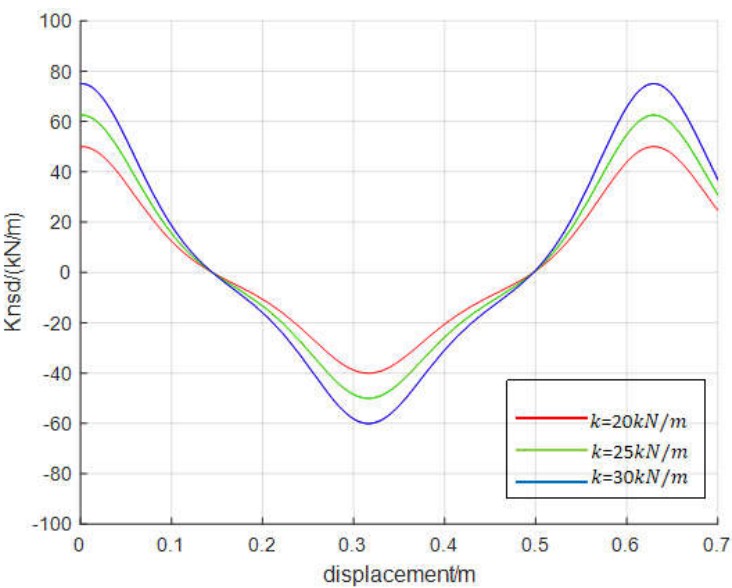

**Figure 10.** Effect of spring stiffness on negative stiffness.

It can be seen from Figure 10 that, in the negative-stiffness stage, the greater the disc spring stiffness *k*, the greater the negative stiffness provided by the device.

### 2.3.3. Influence of Preload $\Delta l$ of Belleville Spring on Negative Stiffness

In accordance with Formula (7), the relationship between negative stiffness, $K_{NSD}$, and displacement, $x$, under different spring preload values, $\Delta l$, is shown in Figure 11.

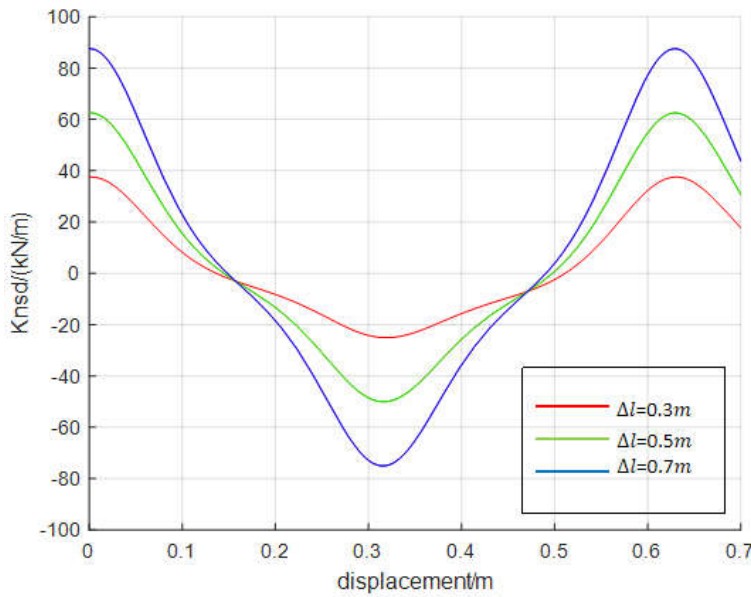

**Figure 11.** Influence of preload of Belleville spring on negative stiffness.

It can be seen from Figure 11 that, the greater the initial compression of the Belleville spring, the greater the negative stiffness provided by the device. However, too much initial compression leads to the reduction of the action range of negative stiffness. It shows that the preloading shrinkage, $\Delta l$, has a significant impact on the negative stiffness of the device.

### 2.3.4. Influence of Friction Coefficient, $\mu$, of Track Surface on Negative Stiffness

In accordance with Formula (7), the relationship between negative stiffness, $K_{NSD}$, and displacement, $x$, at different track surface friction coefficients, $\mu$, is shown in Figure 12.

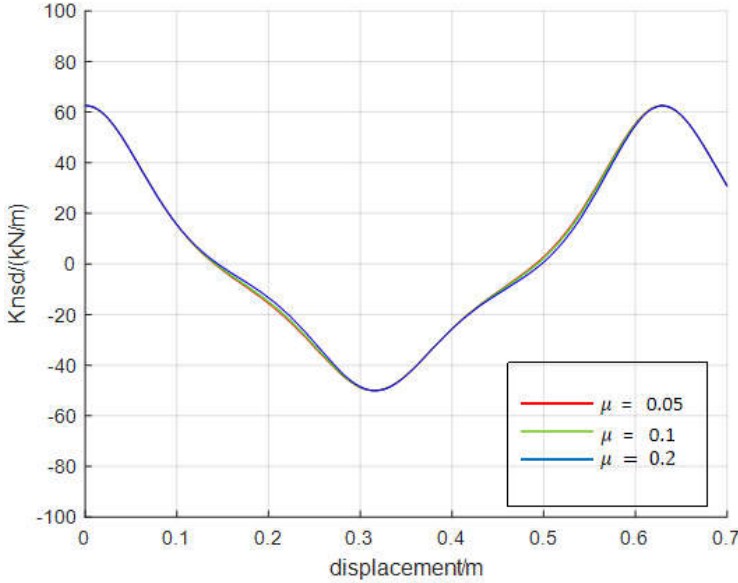

**Figure 12.** Influence of friction coefficient of track surface on negative stiffness.

It can be seen from Figure 12 that changing the friction coefficient, $\mu$, of the track surface has an insignificant effect on the negative stiffness provided by the device.

### 2.4. Parameter Design Method of Negative-Stiffness Device

In order to ensure that the device can effectively provide negative stiffness, the spring should always be in a compressed state; that is, the device should meet the following requirements in the coordinate system shown in Figure 3:

$$\Delta l \geq f(0) - f(\Delta u_{\max}) \tag{12}$$

where $\Delta u_{\max}$ is the maximum value of target control displacement. Assuming that the length of track slab is $L$, the following conditions needs to be met:

$$\Delta u_{\max} \leq \frac{L}{2} \tag{13}$$

$$f'(0) = 0 \tag{14}$$

$$f''(x) < 0(-\Delta u_{\max} \leq x \leq \Delta u_{\max}) \tag{15}$$

## 3. Negative-Stiffness Isolation System and Isolation Principle

The combined isolation system is composed of ordinary lead rubber bearings, belonging to the class of elastomeric bearings [13], and two pre-compression variable-stiffness isolation devices in parallel, and the parallel diagram is shown in Figure 13.

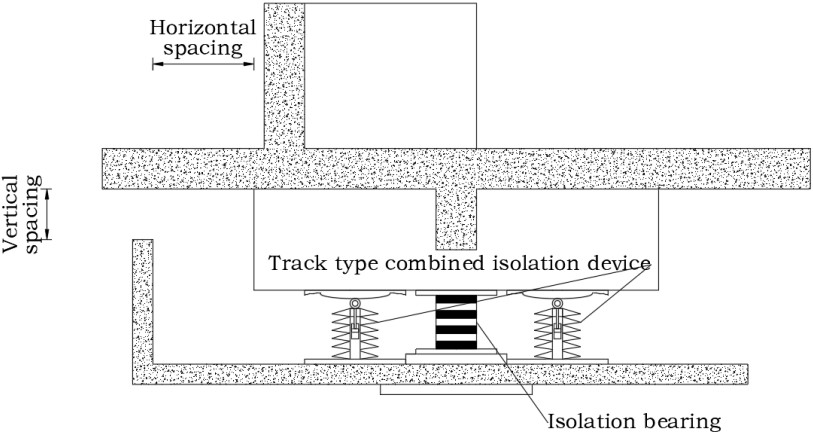

**Figure 13.** Schematic diagram of variable-stiffness vibration isolation device.

The mechanical model of the negative-stiffness isolation system can be obtained by the parallel combination of the mechanical characteristics of the lead rubber bearing and the negative-stiffness device [12], as shown in Figure 14. As can be seen from Figure 14, the isolation system using the negative-stiffness stage of the variable-stiffness device in parallel with the lead rubber bearing cannot only amplify the displacement response of the isolation layer; it can also effectively reduce the seismic response of the superstructure. Therefore, the negative-stiffness part of the isolation composite system is additionally damped to achieve the purpose of reducing the displacement response of the isolation layer and the seismic response of the superstructure at the same time.

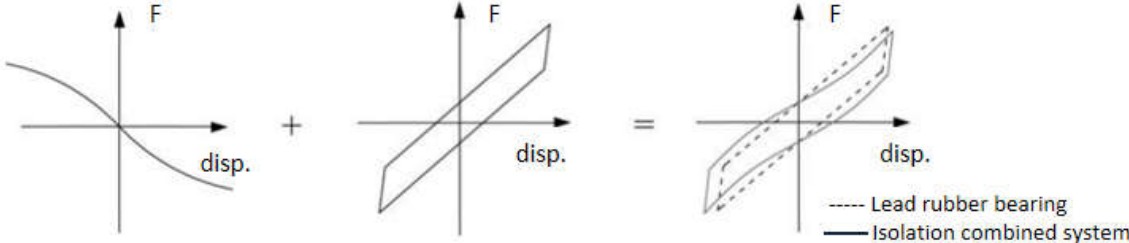

**Figure 14.** Mechanical model of negative-stiffness isolation system.

According to Haringx theory [14,15], the horizontal stiffness of the bilinear model of lead rubber bearing is as follows:

$$K_2 = \frac{p^2}{2K_r q \tan(\frac{qh}{2}) - ph} \tag{16}$$

where $q = \sqrt{p/K_r(1 + p/K_r)}$, $p$ is compression load, $h$ is the total thickness of rubber layer and sandwich steel plate, and $K_r$ is the effective bending stiffness. The initial stiffness, $K_1$ is 11 times that of the horizontal stiffness, $K_2$, so the composite stiffness of the negative-stiffness isolation system is as follows [16]:

Stiffness of the first stage:

$$\begin{aligned} K_1 + K_{NSD} &= \frac{11p^2}{2K_r q \tan(\frac{qh}{2}) - ph} + kf'(x)\left[\frac{f'(x)}{1+f'^2(x)} - \mu\frac{1}{1+f'^2(x)}\right] \\ &+ k[\Delta l + f(x) - f(x_0)]\left[\frac{f''(x)[1+f'^2(x)] - 2f'^2(x)}{[1+f'^2(x)]^2} + \mu\frac{2f'(x)}{[1+f'^2(x)]^2}\right] \end{aligned} \tag{17}$$

Stiffness of the second stage:

$$\begin{aligned} K_2 + K_{NSD} &= \frac{p^2}{2K_r q \tan(\frac{qh}{2}) - ph} + kf'(x)\left[\frac{f'(x)}{1+f'^2(x)} - \mu\frac{1}{1+f'^2(x)}\right] \\ &+ k[\Delta l + f(x) - f(x_0)]\left[\frac{f''(x)[1+f'^2(x)] - 2f'^2(x)}{[1+f'^2(x)]^2} + \mu\frac{2f'(x)}{[1+f'^2(x)]^2}\right] \end{aligned} \tag{18}$$

Considering the additional viscous damper, the dynamic equation of the isolated structure under the combined action of negative stiffness and damping is as follows:

$$m\ddot{u} + (C_1 + C_h)\dot{u} + (k + K_{NSD})u = -m\ddot{u}_g \tag{19}$$

where $m$ is the mass of the isolated structure, $u$ is the displacement of the isolation layer, $C_1$ is the damping coefficient of lead rubber bearing, $C_h$ is the damping coefficient of negative-stiffness device; $k$ is the stiffness of lead rubber bearing, and $K_{NSD}$ is the stiffness of the negative-stiffness device. The natural frequency of the traditional lead rubber bearing isolation structure is $\omega_0 = \sqrt{\frac{k}{m}}$, and the damping ratio of isolation structure is $\xi_0 = \frac{C_1}{2\pi\omega_0}$. The natural circular frequency of the structure with negative stiffness is $\omega = \sqrt{\frac{k+K_{NSD}}{m}} = \omega_0\sqrt{1 + \frac{K_{NSD}}{k}}$. When considering the damping effect of the device, the damping ratio of the isolation structure is as follows [16]:

$$\xi = \frac{C_1 + C_h}{2\pi\omega} = \frac{\xi_0(1 + \frac{C_h}{C_1})}{\sqrt{1 + \frac{K_{NSD}}{k}}} \tag{20}$$

According to the above equation, under the action of the damping negative stiffness, the damping structure self-vibration circle frequency decreases, and the structural damping ratio increases.

## 4. Seismic Response Analysis of Isolated Structure Considering Negative Stiffness

### 4.1. Building Model

The general steps of seismic isolation structure design are shown in Figure 15. The model is a six-story regular reinforced-concrete-frame structure, as shown in Figure 16, with the representative value of gravity load of 35,888.4 kN. The structural isolation layer adopts LRB600. The plane layout of the isolation layer is shown in Figure 17. The product specifications of the isolation rubber bearing are shown in Table 1. The rubber bearings are simulated by the isolator element [17].

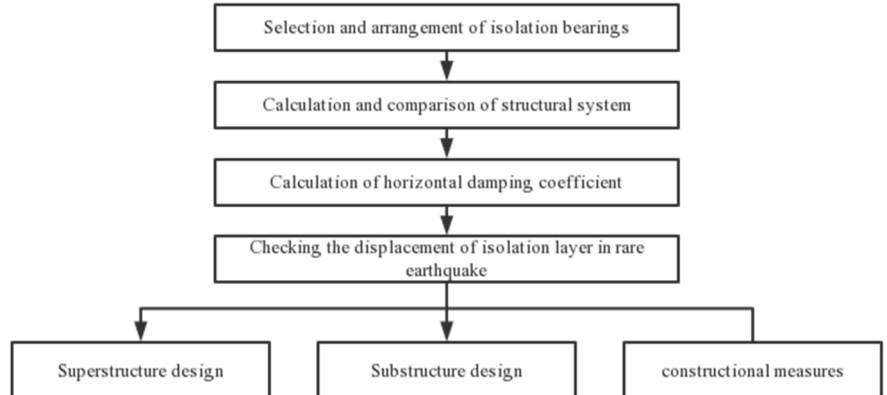

**Figure 15.** Design steps of isolated structure.

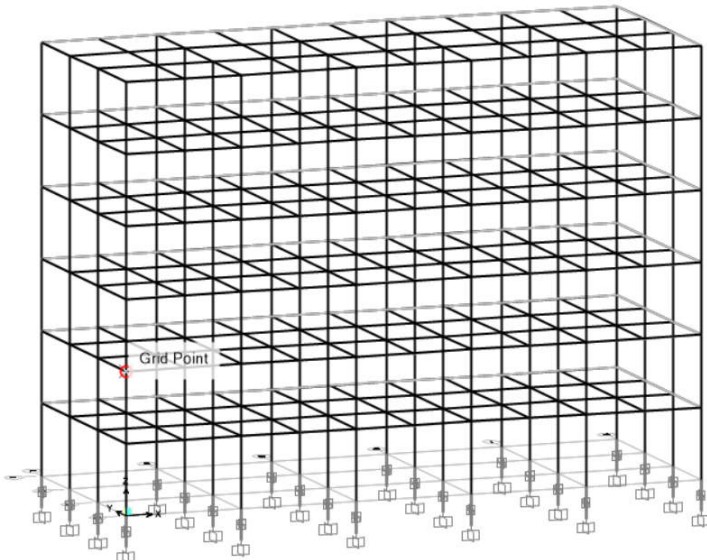

**Figure 16.** Six-story reinforced-concrete-frame structure model.

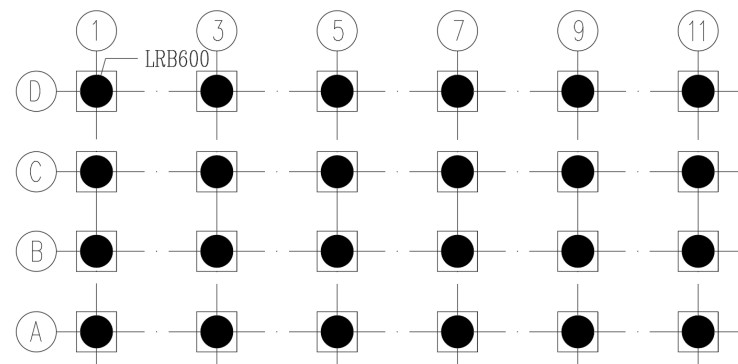

**Figure 17.** Layout of isolation layer.

**Table 1.** Product specification of rubber bearing.

| Model | Effective Diameter | Total Rubber Thickness | Strength Before Yield | Equivalent Stiffness | | Vertical Stiffness | Yield Force |
|---|---|---|---|---|---|---|---|
| | | | | 100% Horizontal Shear Deformation | 250% Horizontal Shear Deformation | | |
| | (mm) | (mm) | kN/m | kN/m | kN/m | kN/mm | kN |
| LRB400 | 400 | 73 | 8790 | 1040 | 820 | 2200 | 27.0 |
| LRB500 | 500 | 92 | 10,910 | 1270 | 1010 | 2400 | 40.0 |
| LRB600 | 600 | 110 | 13,110 | 1580 | 1580 | 2800 | 63.0 |

It is assumed that the structure is located at a region of characteristic period of 0.75 s and seismic intensity of 8; the third group of class IV is adopted for the site [18].

Because the interlayer stiffness of the superstructure is far greater than the horizontal stiffness of the isolation device, and the superstructure only makes horizontal overall translation in the earthquake, in general, in order to simplify the calculation, the superstructure can be simplified as a single-particle system [19], which forms a two-degree-of-freedom model with the isolation layer, as shown in Figure 18. In the figure, $k$ is the total stiffness of the isolation layer, and $c$ is the equivalent damping coefficient of the isolation layer. Using the mechanical model of the negative-stiffness isolation system proposed in this paper, the FNA (Fast Nonlinear Analysis) method is used to analyze the nonlinear time history of the isolation structure [20,21]. In order to ensure the accuracy of the FNA method, the Ritz vector method is used for modal analysis, and the first 50 modes of the structure are considered according to the requirements of accuracy.

The spring part of the negative-stiffness device is simulated by the multi-segment linear elastic element in SAP2000 [22,23]. The mechanical model of the spring part of the negative-stiffness device is simplified into a broken line model (dotted line), as shown in Figure 19. The first stage stiffness takes 0.6 kN/mm, the second stage stiffness takes 0, and the third stage stiffness takes −0.6 kN/mm.

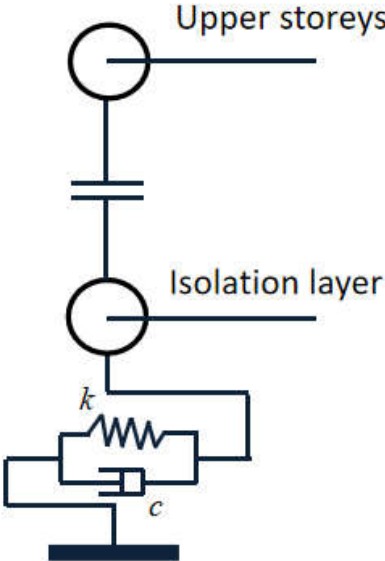

**Figure 18.** Equivalent simplified model of isolated structure.

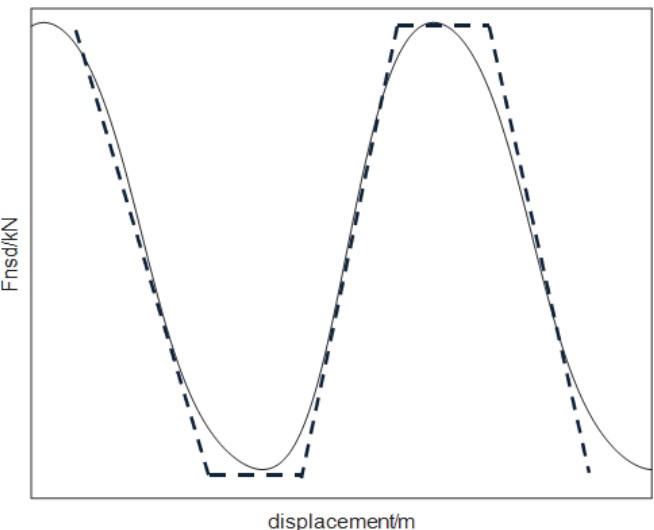

**Figure 19.** Simplified mechanical model of the negative-stiffness device.

### 4.2. Seismic Response

Three natural seismic waves are used for time history analysis: EL Centro, Pasaden, and Santa Barbara Courthouse. The artificial waves synthesized from the isolation response spectrum are REN1 and REN2. The specific information of each natural seismic wave is shown in Table 2, and the response spectrum after peak shaving is shown in Figure 20. The peak value of the seismic wave is taken as 0.3–0.6 g. The models used for analysis are mainly the traditional lead rubber bearing isolation model (LRB), the isolation model with negative stiffness and lead rubber bearing synergy (NSD), and the isolation model with damping negative stiffness and lead rubber bearing synergy (DNSD).

**Table 2.** Seismic-wave recording parameters.

| Earthquake Name | Recording Station | PGA/g |
|---|---|---|
| Imperial Valley-02 | El Centro Array #9 | 0.839 |
| Kern County | Pasadena—CIT Athenaeum | 0.176 |
| Kern County | Santa Barbara Courthouse | 0.322 |
| REN1 | —— | 0.200 |
| REN2 | —— | 0.200 |

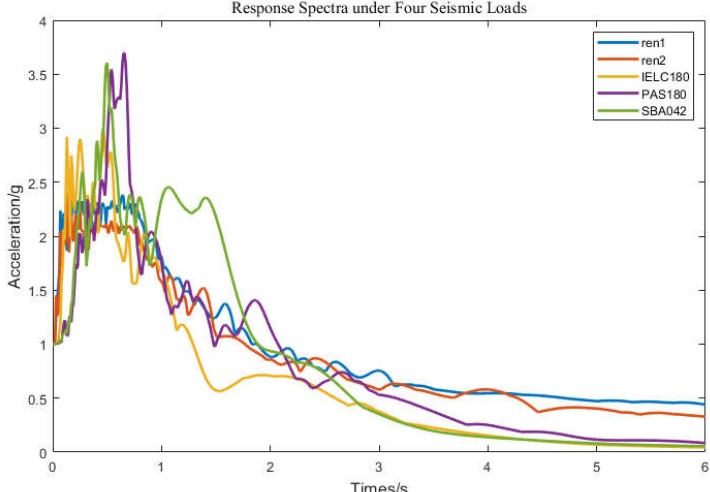

**Figure 20.** Seismic wave response spectrum (after peak shaving).

Table 3 shows the comparison of absolute acceleration response and displacement response results of LRB and DNSD isolation layers under the action of local seismic waves. It can be seen from Table 2 that, when the seismic peak value is 0.3 g, 0.4 g, 0.5 g, and 0.6 g, the acceleration response of DNSD relative to LRB decreases by 14.31%, 10.16%, 14.23%, and 16.20% on average; the displacement response of the isolation layer decreases by 12.44%, 8.72%, 10.17%, and 12.21% on average; and under all working conditions, the acceleration response of DNSD relative to LRB decreases by 13.73% on average, and the displacement response of the isolation layer decreases by 10.89% on average.

**Table 3.** Summary of time history results under the action of local seismic waves (LRB vs. DNSD).

| Seismic Wave | Peak Value of Seismic Wave | Maximum Floor Acc (m/s²) | | | Acc Response Deviation Rate | Maximum Displacement of Isolation Layer (mm) | | | Displacement Response Deviation Rate |
|---|---|---|---|---|---|---|---|---|---|
| | | **LRB** | **NSD** | **DNSD** | | **LRB** | **NSD** | **DNSD** | |
| EL Centro | 0.3 g | 2.01 | 1.90 | 1.58 | −21.39% | 81 | 94 | 65 | −19.75% |
| | 0.4 g | 2.70 | 2.63 | 2.27 | −15.93% | 142 | 152 | 120 | −15.49% |
| | 0.5 g | 3.19 | 2.77 | 2.68 | −15.99% | 207 | 204 | 169 | −18.36% |
| | 0.6 g | 4.27 | 3.62 | 3.13 | −26.70% | 274 | 261 | 220 | −19.71% |
| PAS-180 | 0.3 g | 2.27 | 2.56 | 1.66 | −26.87% | 156 | 160 | 126 | −19.23% |
| | 0.4 g | 3.21 | 2.96 | 2.64 | −17.76% | 226 | 226 | 200 | −11.50% |
| | 0.5 g | 4.15 | 3.42 | 2.98 | −28.19% | 297 | 289 | 261 | −12.12% |
| | 0.6 g | 4.85 | 3.61 | 3.52 | −27.42% | 366 | 347 | 318 | −13.11% |
| SBA−042 | 0.3 g | 2.90 | 3.13 | 2.92 | 0.69% | 175 | 186 | 160 | −8.57% |
| | 0.4 g | 3.85 | 3.82 | 3.43 | −10.91% | 258 | 266 | 237 | −8.14% |
| | 0.5 g | 4.71 | 4.48 | 4.07 | −13.59% | 344 | 339 | 308 | −10.47% |
| | 0.6 g | 5.46 | 5.04 | 4.75 | −13.00% | 429 | 409 | 375 | −12.59% |

**Table 3.** *Cont.*

| Seismic Wave | Peak Value of Seismic Wave | Maximum Floor Acc (m/s²) | | | Acc Response Deviation Rate | Maximum Displacement of Isolation Layer (mm) | | | Displacement Response Deviation Rate |
|---|---|---|---|---|---|---|---|---|---|
| | | LRB | NSD | DNSD | | LRB | NSD | DNSD | |
| REN1 | 0.3 g | 1.43 | 1.35 | 1.29 | −9.79% | 81 | 86 | 75 | −7.41% |
| | 0.4 g | 1.71 | 1.80 | 1.67 | −2.34% | 118 | 132 | 114 | −3.39% |
| | 0.5 g | 2.11 | 2.53 | 2.07 | −1.90% | 156 | 176 | 157 | 0.64% |
| | 0.6 g | 2.74 | 2.90 | 2.87 | 4.74% | 198 | 243 | 189 | −4.55% |
| REN2 | 0.3 g | 1.41 | 1.45 | 1.21 | −14.18% | 69 | 72 | 64 | −7.25% |
| | 0.4 g | 1.80 | 1.90 | 1.73 | −3.89% | 99 | 107 | 94 | −5.05% |
| | 0.5 g | 2.61 | 2.44 | 2.31 | −11.49% | 142 | 160 | 127 | −10.56% |
| | 0.6 g | 3.28 | 2.93 | 2.67 | −18.60% | 189 | 220 | 168 | −11.11% |
| Average response | | | | | −13.73% | | | | −10.89% |

Table 4 shows the comparison of acceleration-response and displacement-response results of LRB and NSD isolation layers under the action of local seismic waves. According to Table 3, when the seismic peak value is 0.3 g, 0.4 g, 0.5 g, and 0.6 g, the acceleration response of NSD relative to LRB decreases by −2.50%, 0.07%, 4.45%, and 10.66% on average; the displacement response of the isolation layer increases by 7.08%, 6.02%, 3.98%, and 4.91% on average; and under all working conditions, the acceleration response of NSD relative to LRB decreases by 3.17% on average, and the displacement response of the isolation layer increases by 5.50% on average.

**Table 4.** Summary of time history results under the action of local seismic waves (LRB vs. NSD).

| Seismic Wave | Peak Value of Seismic Wave | Maximum Floor Acc (m/s²) | | | Acc Response Deviation Rate | Maximum Displacement of Isolation Layer (mm) | | | Displacement Response Deviation Rate |
|---|---|---|---|---|---|---|---|---|---|
| | | LRB | NSD | DNSD | | LRB | NSD | DNSD | |
| EL Centro | 0.3 g | 2.01 | 1.90 | 1.58 | −5.47% | 81 | 94 | 65 | 16.05% |
| | 0.4 g | 2.70 | 2.63 | 2.27 | −2.59% | 142 | 152 | 120 | 7.04% |
| | 0.5 g | 3.19 | 2.77 | 2.68 | −13.17% | 207 | 204 | 169 | −1.45% |
| | 0.6 g | 4.27 | 3.62 | 3.13 | −15.22% | 274 | 261 | 220 | −4.74% |
| PAS-180 | 0.3 g | 2.27 | 2.56 | 1.66 | 12.78% | 156 | 160 | 126 | 2.56% |
| | 0.4 g | 3.21 | 2.96 | 2.64 | −7.79% | 226 | 226 | 200 | 0.00% |
| | 0.5 g | 4.15 | 3.42 | 2.98 | −17.59% | 297 | 289 | 261 | −2.69% |
| | 0.6 g | 4.85 | 3.61 | 3.52 | −25.57% | 366 | 347 | 318 | −5.19% |
| SBA-042 | 0.3 g | 2.90 | 3.13 | 2.92 | 7.93% | 175 | 186 | 160 | 6.29% |
| | 0.4 g | 3.85 | 3.82 | 3.43 | −0.78% | 258 | 266 | 237 | 3.10% |
| | 0.5 g | 4.71 | 4.48 | 4.07 | −4.88% | 344 | 339 | 308 | −1.45% |
| | 0.6 g | 5.46 | 5.04 | 4.75 | −7.69% | 429 | 409 | 375 | −4.66% |
| REN1 | 0.3 g | 1.43 | 1.35 | 1.29 | −5.59% | 81 | 86 | 75 | 6.17% |
| | 0.4 g | 1.71 | 1.80 | 1.67 | 5.26% | 118 | 132 | 114 | 11.86% |
| | 0.5 g | 2.11 | 2.53 | 2.07 | 19.91% | 156 | 176 | 157 | 12.82% |
| | 0.6 g | 2.74 | 2.90 | 2.87 | 5.84% | 198 | 243 | 189 | 22.73% |
| REN2 | 0.3 g | 1.41 | 1.45 | 1.21 | 2.84% | 69 | 72 | 64 | 4.35% |
| | 0.4 g | 1.80 | 1.90 | 1.73 | 5.56% | 99 | 107 | 94 | 8.08% |
| | 0.5 g | 2.61 | 2.44 | 2.31 | −6.51% | 142 | 160 | 127 | 12.68% |
| | 0.6 g | 3.28 | 2.93 | 2.67 | −10.67% | 189 | 220 | 168 | 16.40% |
| Average response | | | | | −3.17% | | | | 5.50% |

Table 5 shows the comparison of acceleration-response and displacement-response results of NSD and DNSD isolation layers under the action of local seismic waves. Table 5 shows that the maximum acceleration response of DNSD is reduced by 10.55% on average compared with NSD. Moreover, the displacement response of the isolation layer decreases by 15.24% on average.

**Table 5.** Summary of time history results under the action of local seismic waves (NSD vs. DNSD).

| Seismic Wave | Peak Value of Seismic Wave | Maximum Floor Acc(m/s²) | | | Acc Response Deviation Rate | Maximum Displacement of Isolation Layer (mm) | | | Displacement Response Deviation Rate |
|---|---|---|---|---|---|---|---|---|---|
| | | LRB | NSD | DNSD | | LRB | NSD | DNSD | |
| EL Centro | 0.3 g | 2.01 | 1.90 | 1.58 | −16.84% | 81 | 94 | 65 | −30.85% |
| | 0.4 g | 2.70 | 2.63 | 2.27 | −13.69% | 142 | 152 | 120 | −21.05% |
| | 0.5 g | 3.19 | 2.77 | 2.68 | −3.25% | 207 | 204 | 169 | −17.16% |
| | 0.6 g | 4.27 | 3.62 | 3.13 | −13.54% | 274 | 261 | 220 | −15.71% |
| PAS-180 | 0.3 g | 2.27 | 2.56 | 1.66 | −35.16% | 156 | 160 | 126 | −21.25% |
| | 0.4 g | 3.21 | 2.96 | 2.64 | −10.81% | 226 | 226 | 200 | −11.50% |
| | 0.5 g | 4.15 | 3.42 | 2.98 | −12.87% | 297 | 289 | 261 | −9.69% |
| | 0.6 g | 4.85 | 3.61 | 3.52 | −2.49% | 366 | 347 | 318 | −8.36% |
| SBA-042 | 0.3 g | 2.90 | 3.13 | 2.92 | −6.71% | 175 | 186 | 160 | −13.98% |
| | 0.4 g | 3.85 | 3.82 | 3.43 | −10.21% | 258 | 266 | 237 | −10.90% |
| | 0.5 g | 4.71 | 4.48 | 4.07 | −9.15% | 344 | 339 | 308 | −9.14% |
| | 0.6 g | 5.46 | 5.04 | 4.75 | −5.75% | 429 | 409 | 375 | −8.31% |
| REN1 | 0.3 g | 1.43 | 1.35 | 1.29 | −4.44% | 81 | 86 | 75 | −12.79% |
| | 0.4 g | 1.71 | 1.80 | 1.67 | −7.22% | 118 | 132 | 114 | −13.64% |
| | 0.5 g | 2.11 | 2.53 | 2.07 | −18.18% | 156 | 176 | 157 | −10.80% |
| | 0.6 g | 2.74 | 2.90 | 2.87 | −1.03% | 198 | 243 | 189 | −22.22% |
| REN2 | 0.3 g | 1.41 | 1.45 | 1.21 | −16.55% | 69 | 72 | 64 | −11.11% |
| | 0.4 g | 1.80 | 1.90 | 1.73 | −8.95% | 99 | 107 | 94 | −12.15% |
| | 0.5 g | 2.61 | 2.44 | 2.31 | −5.33% | 142 | 160 | 127 | −20.63% |
| | 0.6 g | 3.28 | 2.93 | 2.67 | −8.87% | 189 | 220 | 168 | −23.64% |
| Average response | | | | | −10.55% | | | | −15.24% |

It can be seen from the above conclusions that, although the addition of negative-stiffness device can reduce the maximum acceleration response, at the same time, because the displacement of the isolation layer reaches a certain limit, the negative stiffness provided by the negative-stiffness device reduces the total stiffness of the isolation layer and increases the displacement of the isolation layer. After adding the damper to form a composite negative-stiffness device, the maximum acceleration of the floor and the displacement of the isolation layer are significantly reduced, and the energy-dissipation capacity of the isolation layer is enhanced.

The comparison of time history curves of interlayer shear force of LRB, NSD, and DNSD under the 0.6 g peak input of five seismic waves is shown in Figure 21. It can be seen from Figure 21 that the bottom shear response of NSD is the smallest and the LRB is the largest, indicating that it is effective to further reduce the seismic response of the superstructure with negative stiffness. Figure 22 shows the comparison of displacement angles between the lower layers of LRB, NSD, and DNSD at the 0.5 g peak input of five seismic waves. It can be seen from Figure 22 that there is little difference between the interlayer displacement angles of NSD and DNSD, but they are significantly reduced relative to LRB, indicating that both NSD and DNSD can reduce the seismic response of the superstructure, and for reducing the interlayer displacement angle, the negative stiffness plays a controlling role compared with the additional damping.

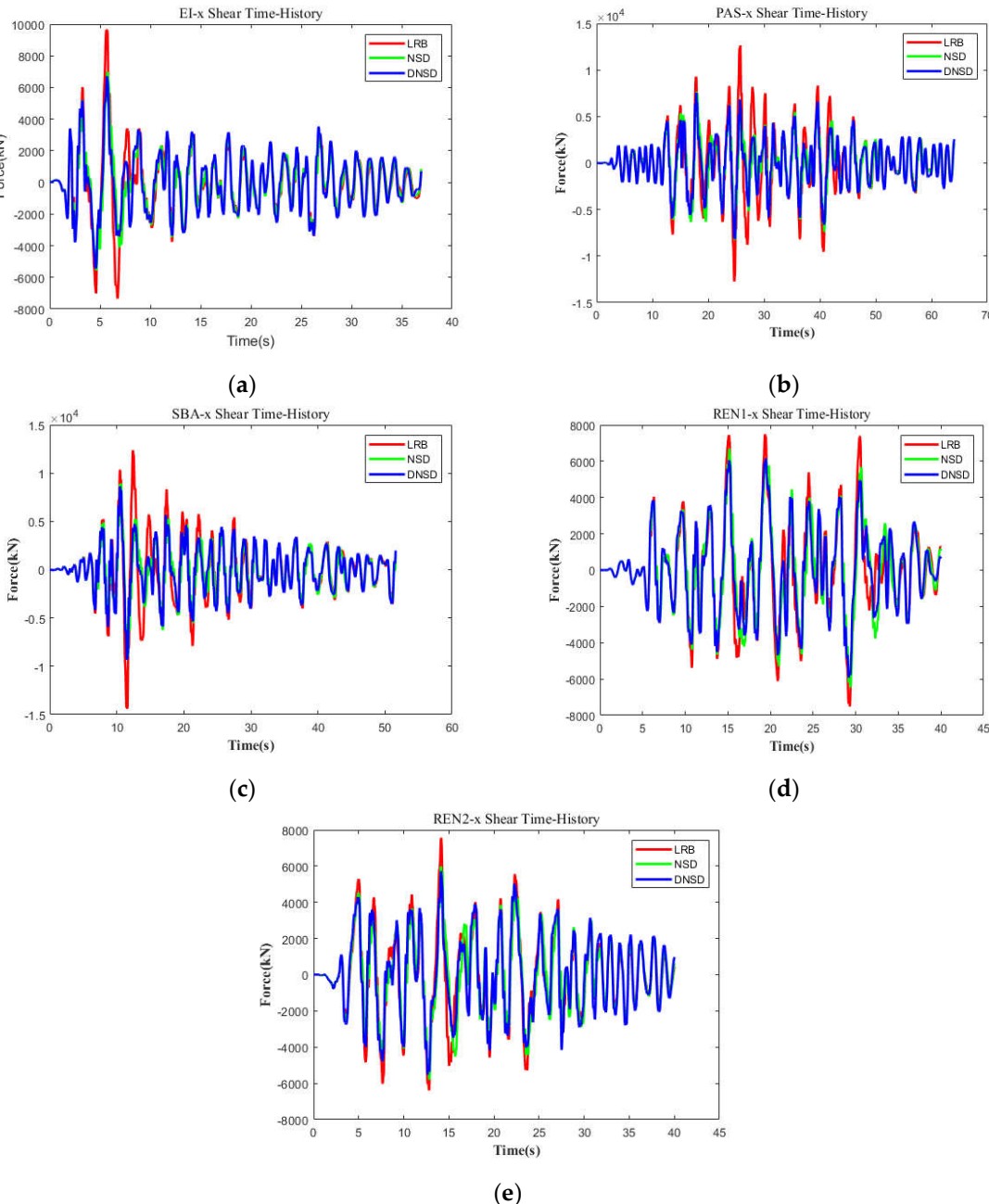

**Figure 21.** Time-history comparison of bottom shear response among three models under different earthquake waves at a peak of 0.6 g. (**a**) Time history of shear response of ground floor under 0.6 g peak of EL wave. (**b**) Time history of shear response of ground floor under 0.6 g peak of PAS wave. (**c**) Time history of shear response of ground floor under 0.6 g peak of SBA wave. (**d**) Time history of shear response of ground floor under 0.6 g peak of REN1 wave. (**e**) Time history of shear response of ground floor under 0.6 g peak of REN2 wave.

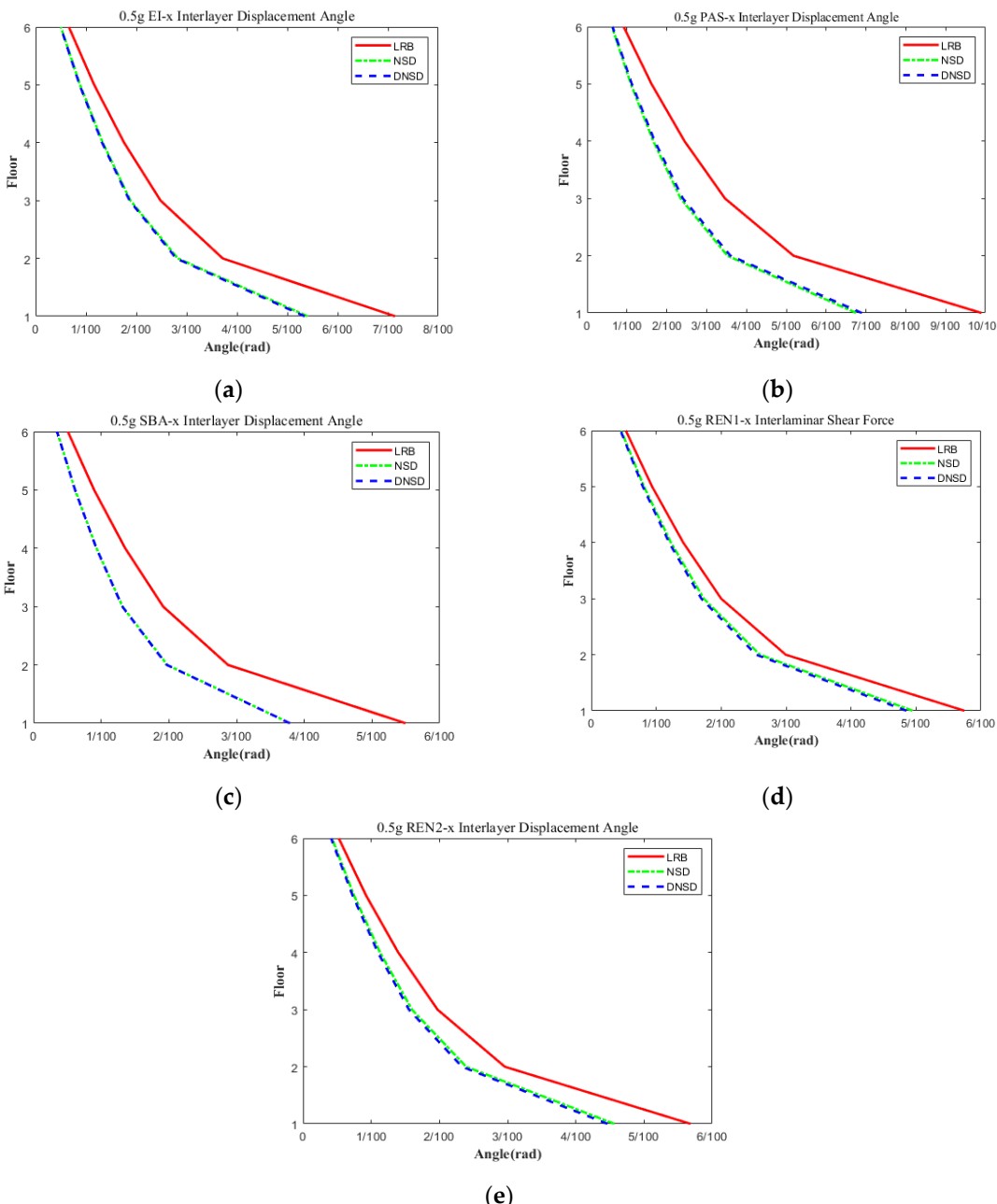

**Figure 22.** Comparison of inter-story displacement angles among three models under a 0.5 g peak input of different seismic waves. (**a**) Interlaminar displacement angle of 0.5 g peak value of EL wave. (**b**) Interlaminar displacement angle of 0.5 g peak value of PAS wave. (**c**) Interlaminar displacement angle of 0.5 g peak value of SBA wave. (**d**) Interlaminar displacement angle of 0.5 g peak value of REN1 wave. (**e**) Interlaminar displacement angle of 0.5 g peak value of REN2 wave.

The comparison of floor accelerations of LRB, NSD, and DNSD under the peak inputs of 0.3 g and 0.6 g of five seismic waves is shown in Figures 23 and 24, respectively. It can be seen from Figure 23 that, under the seismic wave input with a small peak value, the negative stiffness can still play a certain role in reducing the floor acceleration. However, due to the small displacement of the isolation layer at this time, the negative stiffness does not fully play its role, and the acceleration of the upper floor may increase compared with the LRB. Compared with Figure 24, under the input of large peak seismic wave, we can see that the floor acceleration of NSD and DNSD is significantly lower than that of LRB, and the negative stiffness plays a fuller role. Under this working condition, the curves of NSD

and DNSD almost coincide, showing that the control factor for reducing the response of the upper floor is negative stiffness.

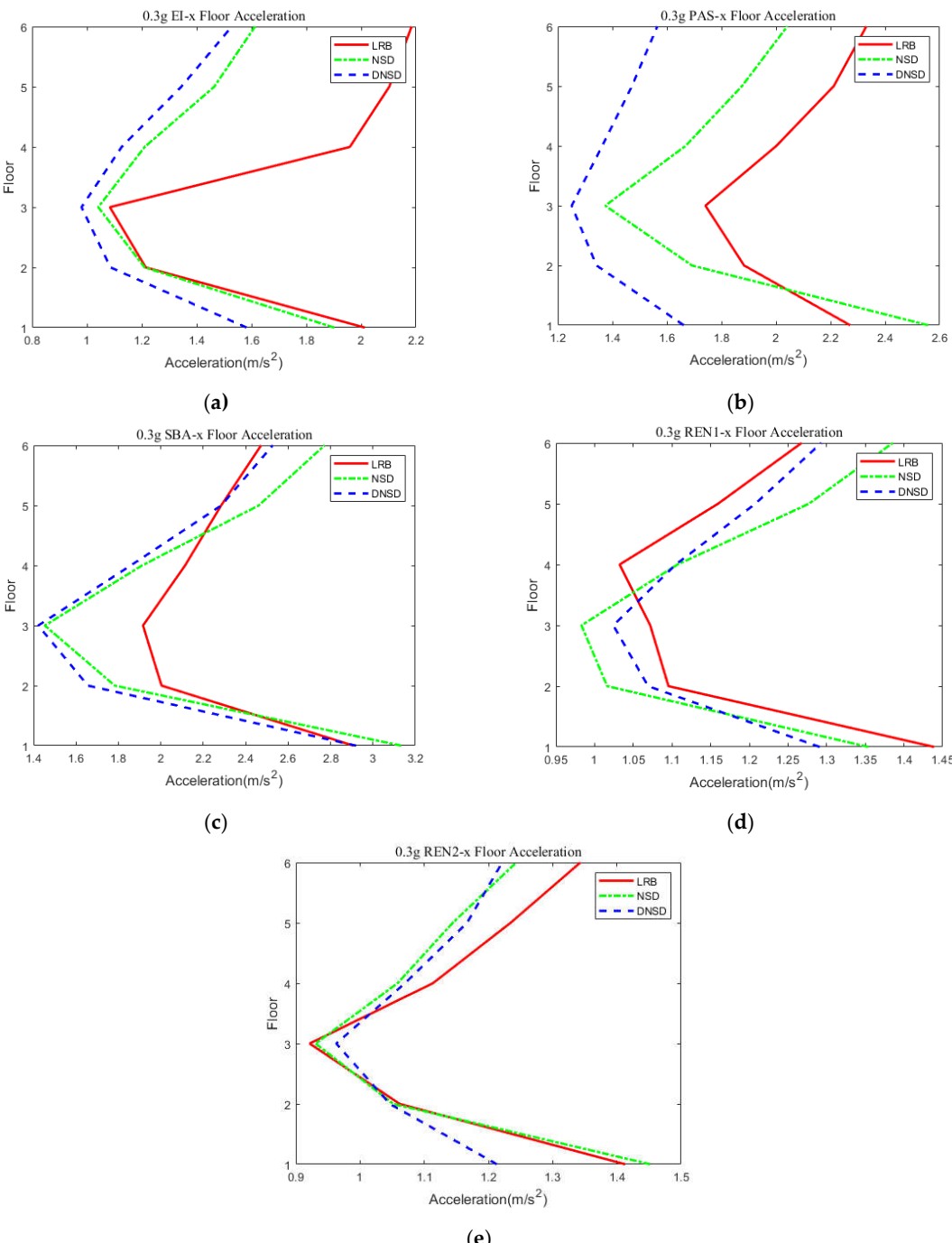

**Figure 23.** Comparison of floor accelerations among three models under different earthquake waves at a peak of 0.3 g. (**a**) Floor acceleration at 0.3 g peak of El wave (**b**) Floor acceleration at 0.3 g peak of PAS wave. (**c**) Floor acceleration at 0.3 g peak of SBA wave. (**d**) Floor acceleration at 0.3 g peak of REN1 wave. (**e**) Floor acceleration at 0.3 g peak of REN2 wave.

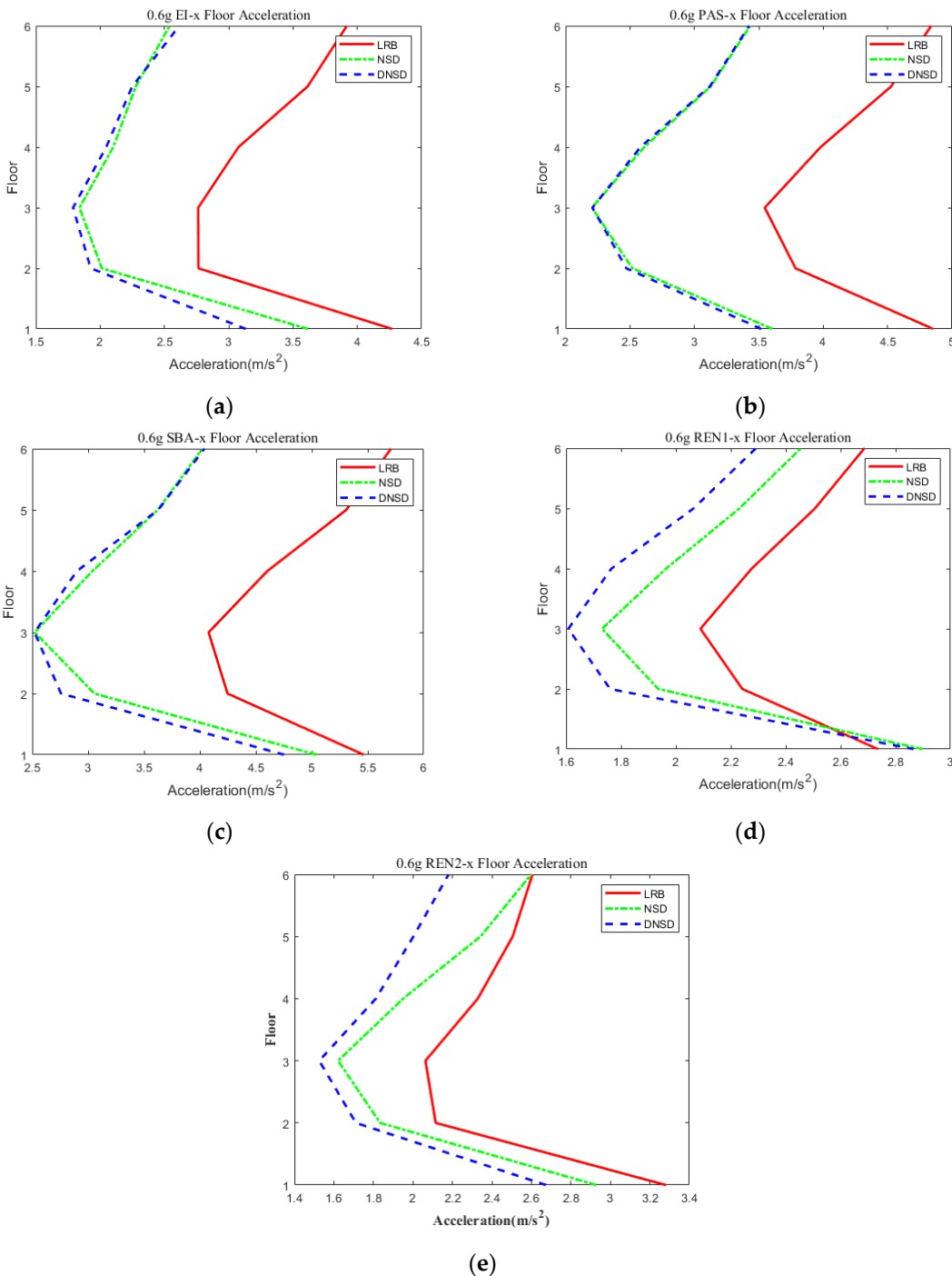

**Figure 24.** Comparison of floor accelerations among three models under different earthquake waves at a peak of 0.6 g. (**a**) Floor acceleration at 0.6 g peak of El wave. (**b**) Floor acceleration at 0.6 g peak of PAS wave. (**c**) Floor acceleration at 0.6 g peak of SBA wave. (**d**) Floor acceleration at 0.6 g peak of REN1 wave. (**e**) Floor acceleration at 0.6 g peak of REN2 wave.

Figure 25 shows the comparison of hysteretic curves of the isolation layer of LRB, NSD, and DNSD under the 0.6 g peak input of five seismic waves. It can be seen from Figure 25 that the displacement of the DNSD model considering damping is significantly reduced, and the displacement control effect is better, while the displacement of NSD may slightly increase compared with LRB. It is verified that under the action of other small seismic peaks, the hysteretic curve characteristics of the isolation layer are still consistent with the above conclusions.

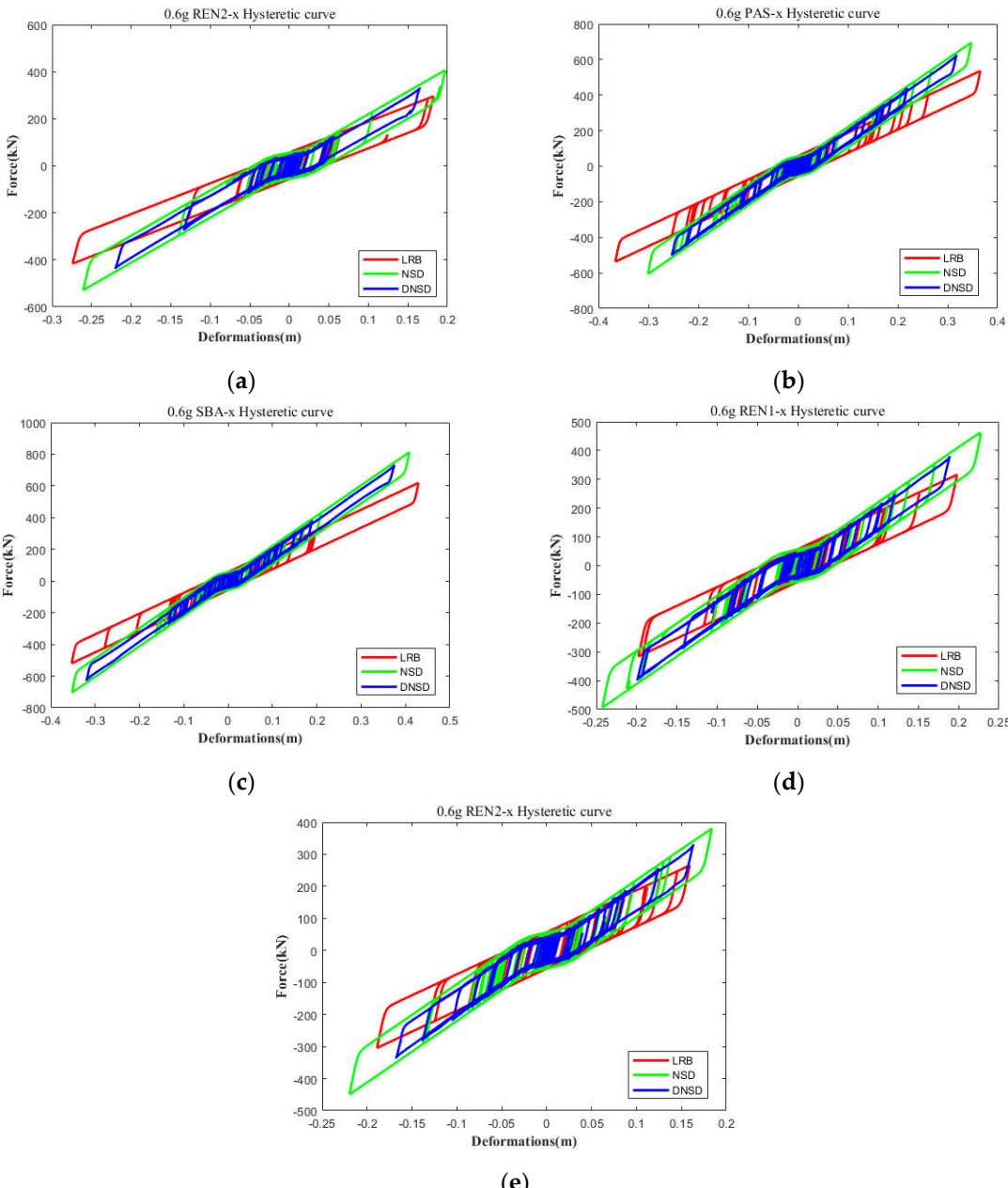

**Figure 25.** Comparison of hysteretic curves among three models under different earthquake waves at a peak of 0.6 g. (**a**) Hysteresis loops under EL seismic wave at peak of 0.6 g. (**b**) Hysteresis loops under PAS seismic wave at peak of 0.6 g. (**c**) Hysteresis loops under SBA seismic wave at peak of 0.6 g. (**d**) Hysteresis loops under REN1 seismic wave at peak of 0.6 g. (**e**) Hysteresis loops under REN2 seismic wave at peak of 0.6 g.

The comparison of the hysteretic curves of the viscous dampers of DNSD under the peak input of 0.3 g and 0.6 g of five seismic waves is shown in Figures 26 and 27, respectively. It can be seen from Figures 26 and 27 that, under large earthquakes, the hysteretic curve of the damper is fuller, thus making it possible to dissipate more seismic energy and reduce the energy input to the superstructure.

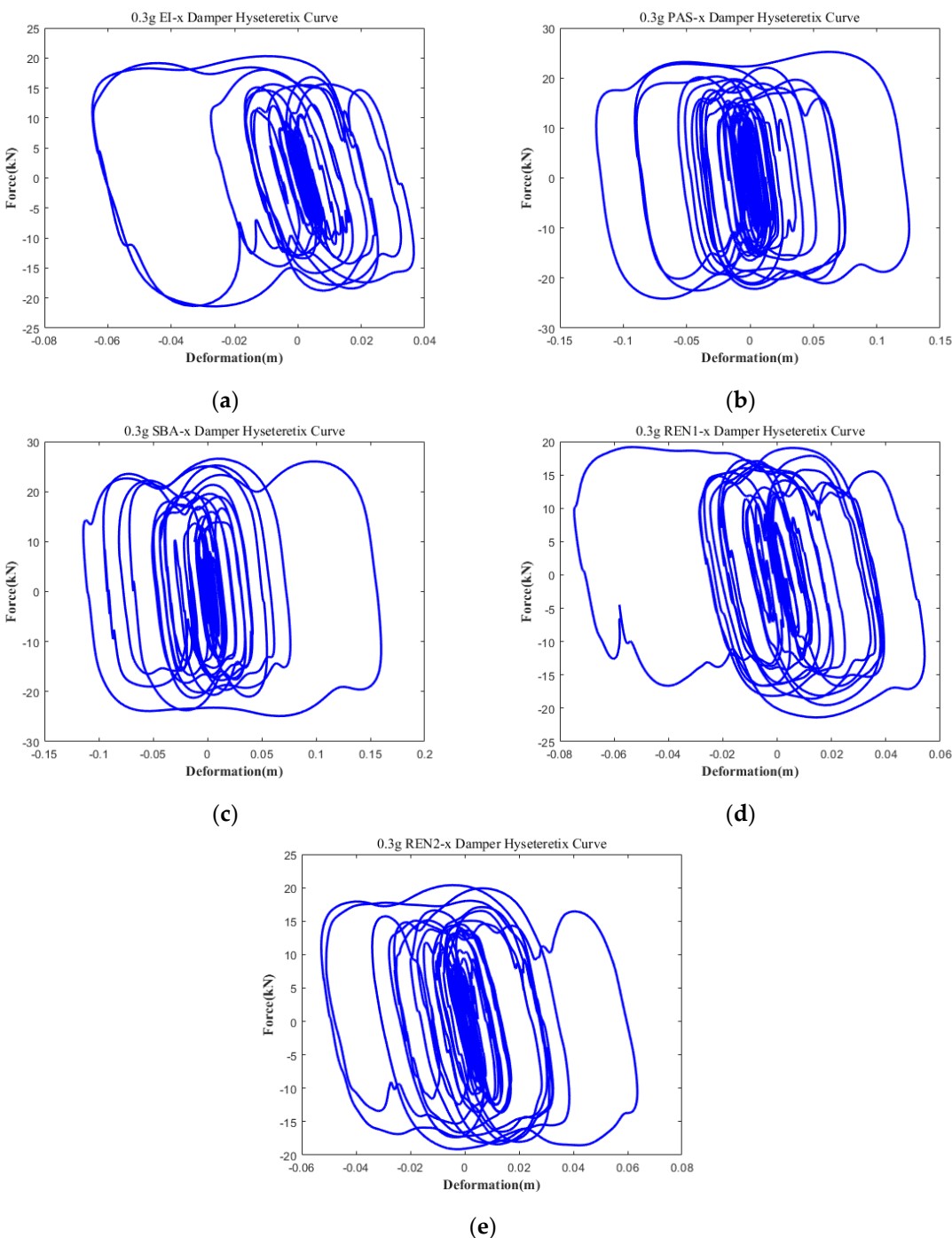

**Figure 26.** Hysteretic curves of viscous dampers under different earthquake waves at a peak of 0.3 g. (**a**) Hysteresis loops under EL seismic wave at peak of 0.3 g. (**b**) Hysteresis loops under PAS seismic wave at peak of 0.3 g. (**c**) Hysteresis loops under SBA seismic wave at peak of 0.3 g. (**d**) Hysteresis loops under REN1 seismic wave at peak of 0.3 g. (**e**) Hysteresis loops under REN2 seismic wave at peak of 0.3 g.

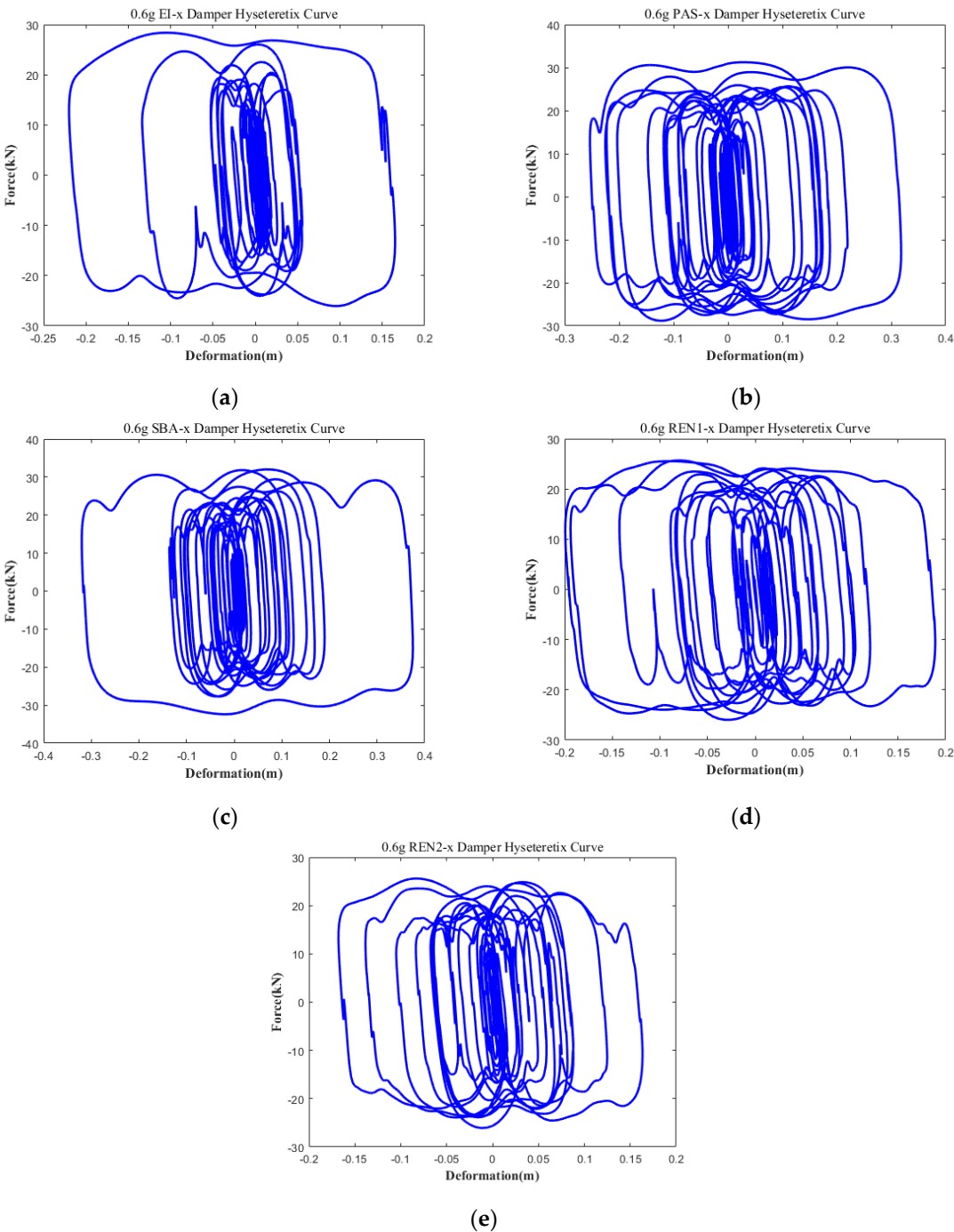

**Figure 27.** Hysteretic curves of viscous dampers under different earthquake waves at a peak of 0.6 g. (**a**) Hysteresis loops under EL seismic wave at peak of 0.6 g. (**b**) Hysteresis loops under PAS seismic wave at peak of 0.6 g. (**c**) Hysteresis loops under SBA seismic wave at peak of 0.6 g. (**d**) Hysteresis loops under REN1 seismic wave at peak of 0.6 g. (**e**) Hysteresis loops under REN2 seismic wave at peak of 0.6 g.

## 5. Conclusions

Improving the isolation effect and displacement response of the isolation layer of long-period isolated structures is an urgent problem to be studied and solved at present. In this paper, a negative-stiffness device was proposed and analyzed theoretically, its mechanical model was established, and the seismic response of its isolation structure system was

analyzed and studied. The seismic responses of the LRB, NSD, and DNSD models were compared. The main conclusions are as follows:

(1) A negative-stiffness device composed of preloaded Belleville spring, guide sleeve, one-way roller, track surface, and viscous damper was proposed. The device uses a preloaded Belleville spring and track surface to provide a restoring force consistent with the direction of displacement, so as to realize the negative-stiffness effect. Additional damping enhances the energy dissipation capacity of the device and controls the displacement of the isolation layer, so as to reduce the displacement response of the isolation layer and the ground motion response of the superstructure at the same time.

(2) The mechanical model of the negative-stiffness device was established, and the influence of various parameters on the mechanical model of the negative-stiffness device was analyzed.

(3) The time history analysis results show that, under all working conditions, the acceleration response of NSD relative to LRB decreases by 3.17% on average, and the displacement response of the isolation layer increases by 5.50%. Under all working conditions, the acceleration response of DNSD relative to LRB decreases by 13.73% on average, and the displacement response of the isolation layer decreases by 10.89% on average. Under all working conditions, the isolation effect of NSD and DNSD is superior to that of LRB model, and the greater the peak value of input seismic wave, the more obvious the isolation effect is. It does not only realize the effective isolation of long-period structures but also improves the disadvantages of the previous negative-stiffness devices, such as their complex structure and residual deformation. Under the action of load, the device has good deformation capacity and can maintain stable small stiffness change, thus further ensuring the stability and reliability of the combined use of isolation bearings.

(4) In this paper, the track surface function was studied only as the cosine function, but in fact, there are countless kinds of track surface functions that meet the boundary conditions in theory. How to adjust the track surface function to make the isolation performance better is still a problem that needs to be solved by a lot of research. At the same time, the track surface can be made into a functional surface in both $x$ and $y$ directions, so that the building has good seismic isolation performance in different directions.

(5) In the future, the negative-stiffness model device will be designed and manufactured, and the mechanical model of the device will be obtained through the static mechanical performance test, which is compared with the theoretical analysis results; and the dynamic test research of the negative-stiffness isolation device and isolation system will be further carried out.

**Author Contributions:** Conceptualization, Y.Y.; methodology, G.K. and Y.Y.; software, G.K. and Z.W.; formal analysis, G.K.; resources, Y.Y.; data curation, G.K. and Z.W.; writing—original draft preparation, G.K.; writing—review and editing, G.K., Z.W. and Y.Y.; supervision, Y.Y.; project administration, Y.Y. All authors have read and agreed to the published version of the manuscript.

**Funding:** This research received no external funding.

**Institutional Review Board Statement:** Not applicable.

**Informed Consent Statement:** Not applicable.

**Data Availability Statement:** Not applicable.

**Conflicts of Interest:** The authors declare no conflict of interest.

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
