# Peer review of "Mechanical Model and Seismic Response Analysis of a Track Type Combined Isolation Device"

_applsci, doi:10.3390/app12178794_

Round 1

Reviewer 1 Report

The manuscript presents an interesting numerical investigation concerning a mechanical model and seismic response analysis of a track type combined isolation device. The topic is relevant and the reviewer believes that the manuscript's subject is within the journal's scope However, some aspects described below must be better clarified and discussed. Also, the revision and improvement of the english language shall be carried out. Based on the following list of observations/suggestions, I recommend a major revision of the manuscript.

Abstract

·        Please revise lines 11-20. The idea is not pretty straightforward and the language needs deep revision.

Section 1

Other strategies for global structure retrofitting shall be presented, e.g. steel braces, or RC shear walls, or retrofitting of elements. Please consider the following research examples (e.g. Doi:         10.1016/j.istruc.2015.03.001; 10.1016/j.engstruct.2017.05.021)

·        The novelty of this work needs to be well presented.

Section 2

·        Please check in the entire manuscript if the variables are well introduced right after presenting the equations;

·        The discussion concerning the use of different β is not well presented. Why does it provoke such modification in the force-displacement curves?

Section 3

·        Please re-check the formatting of Equations 17 and 18

Section 4

·        This section is slightly poor. The authors provide very few details, e.g. the isolation layer is not well introduced and explained. Please revise it.

·        Figure 17: Please improve the Figure quality.

Section 5

·        Please include future works and identify the existing gaps in the literature

Reviewer 2 Report

The paper needs to be reconsidered for publication after the authors addressing ALL the issues highlighted in the pdf file.

Author Response

Replies have been made at all highlighted parts. Please see the attachment.

Round 2

Reviewer 1 Report

The authors carried out almost all the revisions and recommendations made by the reviewers. The manuscript is now suitable for publication.

Reviewer 2 Report

The authors made the required corrections thus the paper can be accepted for publication.